# OpenReview forum: "Causally Motivated Sycophancy Mitigation for Large Language Models"
_ICLR.cc/2025/Conference — ICLR 2025 Poster_

### Official Review · Reviewer_Zk8b · 2024-10-21

**Soundness:** 3
**Presentation:** 2
**Contribution:** 3
**Rating:** 6
**Confidence:** 4

**Summary:**

This paper analyzes and models sycophancy in LLMs through the lens of structured causal models (SCMs), which is actually the reliance on spurious correlations between user preferences and model outputs. Based on the proposed SCMs, this paper develops a novel framework called CAUSM to mitigate sycophancy in LLMs by exploiting a significant signature.

**Strengths:**

1.The author models the phenomenon of sycophancy in language models as a type of spurious correlation in causal structures, making it possible to address sycophancy through conditional independence constraints.
2.The CAUSM method achieves excellent results in mitigating sycophancy across INTRA-DATASET, CROSS-DATASET, and CROSS-TASK scenarios.

**Weaknesses:**

1. The motivation for the Causal Activation Calibration method in Section 4.3 is unclear. Specifically, the relationship between the causal direction in Section 4.3 and the SCM in Section 4.1 is not sufficiently clear. Please provide further clarification on this point. My understanding is that Equation (6) aims to add a conditional independence constraint to the original training objective, intending to eliminate spurious correlations between $Z_S$  and $Y$. However, I am not convinced how the “causal direction” in Section 4.3 effectively mitigates these spurious correlations. Does this approach leverage the causal direction as a representation of causal effect orientation, or is there some other theoretical justification? I suggest that the authors clarify this aspect in the paper.

2. The second statement in Lemma 4.1 is not described clearly enough. I would like to understand which part of the subsequent methods specifically utilizes this statement. For instance, if we consider a specific example where $f(Z_C,Z_S) = Z_C$, then this condition does not hold in the causal graph shown in Figure 2(a). Furthermore, does the second statement in Lemma 4.1 offer any guidance in constructing the subsequent methods? If not, I would suggest the authors consider removing this statement.

**Questions:**

1. Why does accuracy decrease as K increases, and what factors influence the optimal K value?
2. How does the paper measure and rank the sycophancy-relatedness of all attention heads and select the top-K heads?
3. What are the strengths and weaknesses of Sycophancy Heads Pruning and Sycophancy Representation Editing, and are there any practical suggestions for choosing between them?

**Details Of Ethics Concerns:**

I have no ethics concerns regarding this paper.

---

### Official Review · Reviewer_MyvT · 2024-11-04

**Soundness:** 2
**Presentation:** 2
**Contribution:** 3
**Rating:** 6
**Confidence:** 3

**Summary:**

This paper addresses the issue of sycophancy in large language models (LLMs) and introduces CAUSM, a novel method for identifying and mitigating sycophantic behavior within the models’ latent representations. The authors view sycophancy in LLMs as “spurious correlations between user preferences and model outputs”. By leveraging structured causal models, they aim to disentangle sycophantic representations from causal embeddings. An intervention-based technique is then developed to recalibrate the causal representation direction embedded in attention heads.

**Strengths:**

1. The paper aims to address LLMs’ sycophancy issue, which is an important topic in the community.

2. A variety of experiments have been conducted to show the effectiveness of the proposed approach across different datasets.

3. The high-level structure of the paper is easy to follow.

**Weaknesses:**

1. The motivation for the proposed approach and the intuition of the algorithm design needs to be more clear.
-  In the Introduction, the authors discuss two groups of prior research on LLM sycophancy: (1) linear probing and (2) path patching. However, these prior works primarily concentrate on analyzing and understanding sycophantic behavior in LLMs, rather than on mitigating it (also as mentioned in Related Work line 113-122). Given the distinct emphasis of these studies compared to the authors' goal, it is unclear how the limitations of these earlier works directly motivate the development of the authors’ proposed approach.

- While the authors discuss several recent studies on mitigating LLM sycophancy in the Related Work section (lines 124–135), a more in-depth comparison between these studies and their own approach would be beneficial. Specifically, it would be helpful to understand if there are potential methodological concerns with the designs proposed by Burns et al. (2022) and Rimsky et al. (2023) that inspired the development of CAUSM.

- It would be helpful to discuss the rationale for using structured causal models to capture sycophancy in the models’ latent representations.

2. Some compared methods in the tables/figures of results are confusing. E.g., I couldn’t find a clear definition of CAUSM (Base) (Table 1 & 2) and CAUSM (Table 2). What is the difference between CAUSM and CAUSM (Base) in Table 2?

3. Did the authors conduct the experiments on multiple base LLMs or specifically focus on a single base LLM (i.e., Llama-2-7B-Chat)?

**Questions:**

Please see the section on Weaknesses.

---

### Official Review · Reviewer_FC8R · 2024-11-04

**Soundness:** 2
**Presentation:** 2
**Contribution:** 2
**Rating:** 6
**Confidence:** 2

**Summary:**

The paper introduces a novel framework called CAUSM to address sycophancy in large language models (LLMs), which refers to the models’ tendency to align with user preferences even when those preferences lead to incorrect or biased outputs. This behaviour reduces the reliability and factual integrity of LLM responses.  The authors conclude that CAUSM effectively mitigates sycophantic behaviour in LLMs by focusing on the causal structure of sycophantic representations. The framework offers a scalable solution to improve the factual reliability of LLMs while respecting user preferences, which holds promise for enhancing trust in AI outputs in real-world applications.

**Strengths:**

1. The authors provide a novel framework, CAUSM, which leverages structured causal models to address sycophancy in large language models (LLMs). By introducing a causal approach, they advance beyond existing methods that may depend on spurious correlations, achieving more reliable mitigation of sycophantic responses.

2. The article is well-structured, with clearly delineated sections detailing the problem (sycophancy in LLMs), prior approaches, and the limitations they aim to address with CAUSM. This clarity is helpful for readers who may be less familiar with the subject matter.

**Weaknesses:**

My main concerns focus on the application of the structured causal model (SCM) approach. The authors state that causal relations can be captured via a directed acyclic graph (DAG) learned through regularization. However, in real-world applications, the regularization term is unlikely to reach zero. How can the authors be certain that the learned representations are truly disentangled? The methodology would benefit from a more robust approach to verifying that these representations capture causal rather than correlated information.

My second question relates to the evaluation of DAG structures in SCMs. Typically, metrics like Structural Hamming Distance (SHD) or False Discovery Rate (FDR) are used to confirm if the learned graph conforms to a DAG structure. This paper, however, claims that the approach is inspired by graph models without providing a specific graph-based evaluation. How can we be sure that the learned representations are genuinely causality-related rather than optimized merely by overfitting through additional parameters? Both theoretical analysis and case studies would strengthen the authors’ claims.

Regarding the representation learning approach, it seems counterintuitive that a fully supervised learning model without any stochastic components could infer causality from data alone. The mutual information-based independence criterion here could indicate correlation, but correlation does not imply causation. Causal inference methods typically rely on stochasticity in at least one part of the model (e.g., a two-tower architecture) to differentiate causality from mere correlation. Without this, there is a risk of learning coincidental patterns rather than true causal relationships.

Lastly, a minor issue: without providing a formal proof, I recommend avoiding the use of terms like "Lemma" to present conclusions, as this suggests a level of mathematical rigor that is not fully substantiated in the paper.

**Questions:**

See in weaknesses.

The author does not need to address each identified weakness individually. Instead, it would be more effective to provide additional experimental results that demonstrate that the learned representations truly capture causality, beyond merely improving performance. For instance, could the correlation between the generated text and the learned representation be measured? Does the disentangled representation contribute more meaningfully to sycophancy-related terms or phrases? A more fine-grained case study focused on these aspects would offer stronger evidence for the proposed approach than the current, label-focused explanations.

I would like to increase my score if the author could provide more detailed results.

---

### Official Review · Reviewer_bq5H · 2024-11-04

**Soundness:** 2
**Presentation:** 2
**Contribution:** 2
**Rating:** 6
**Confidence:** 4

**Summary:**

This paper introduces a new framework called CAUSM (Causally Motivated Sycophaocy Mitigation) aimed at reducing sycophancy in Large Language Models (LLMs). The paper analyzes and models the sycophancy issue in LLMs through the lens of Structured Causal Models (SCMs).
A significant causal signature is proposed to distinguish latent causal embeddings from spurious embeddings that cause sycophancy. The paper further propose an intervention-based scheme to calibrate the direction of the derived causal representations. Extensive experiments show that the proposed approaches outperforms the state-of-the-art competitors.

**Strengths:**

1. It is the first to apply Structured Causal Models (SCMs) to analyze and model sycophancy behavior in Large Language Models (LLMs), offering an innovative research perspective.
2. Extensive experiments show that CAUSM is superior to existing state-of-the-art methods in mitigating sycophancy in LLMs.

**Weaknesses:**

1. In Line 81, the phrase “To map the latent causal embeddings to the observable intermediate components of LLMs” appears to conflict with the statement “a significant causal signature which can distinguish the intended causal embeddings from spurious embeddings which incur sycophancy within the latent representation space.” Could you clarify this discrepancy?

2. How can I(X_P ; Y | Z) be approximated by Eq. 7? Is causal intervention controllable? Could you provide an example to illustrate this?

3. Why does the intervention \bar{X}_P maximize the difference in cross-entropy losses for a fixed W? Is this a result of the algorithm's design, or does it align with the intrinsic nature of interventions?

4. The author claims to utilize Parameter-Efficient Fine-Tuning (PEFT) in Line 267, yet states in the baselines section that all parameters are fine-tuned. This is rather confusing.

5. In Section 4.3, how is the weight matrix value obtained, and what does |w_l^h| represent? Are these parameters part of the adaptor in PEFT, or are they parameters of the model itself? How are these parameters utilized specifically?

**Questions:**

Typo:
Line 200, remove "a directed"

---

### Meta-Review · Area_Chair_PhA4 · 2024-12-14

**Metareview:**

The paper addresses sycophancy in large language models.

Strengths:
The first to apply Structured Causal Models (SCMs) to analyze and model sycophancy
Extensive experimentations

Weaknesses:
Some issues such as clarity might have been addressed in the rebuttal, still it might be worth improving it for the camera ready.
See proposals to improve the paper by different reviewers.
Well written,

Overall, the paper addresses an interesting and well explained issue.

**Additional Comments On Reviewer Discussion:**

There was significant effort in the rebuttal that might improve the paper further, as it was unanimously agreed this should be accepted it didn't matter to the decision and should mainly be used to make the camera ready of the paper better.

---

### Decision · Program_Chairs · 2025-01-22

Accept (Poster)